# Long Title: Cas10 based 7SL-sRNA diagnostic for the detection of active trypanosomosis

**Sabine Grüschow**[1]☯, **Pieter C. Steketee**[2]☯, **Edith Paxton**[2], **Keith R. Matthews**[3], **Liam J. Morrison**[2], **Malcolm F. White**[1]\*, **Finn Grey** (ID)[2]\*

1 School of Biology, University of St Andrews, St Andrews, United Kingdom, 2 The Roslin Institute, Royal (Dick) School of Veterinary Studies, University of Edinburgh, Edinburgh, United Kingdom, 3 Institute of Immunology and Infection, School of Biological Sciences, University of Edinburgh, Edinburgh, United Kingdom

☯ These authors contributed equally to this work
\* mfw2@st-andrews.ac.uk (MFW); fgrey@ed.ac.uk (FG))

## Abstract

Animal Trypanosomosis (AT) is a significant disease affecting cattle across sub-Saharan Africa, Latin America, and Asia, posing a major threat to economic productivity and animal welfare. The absence of reliable diagnostic tests has led to an over-reliance on widespread pre-emptive drug treatments, which not only compromise animal health but also heighten the risk of drug resistance. The chronic nature of AT, characterized by cyclical low or undetectable parasite levels, and the necessity for field-applicable tests that can distinguish between active infection and prior exposure, present considerable challenges in developing effective diagnostics. In previous work, we identified a parasite-specific small RNA, 7SL-sRNA, which is detectable in the serum of infected cattle, even during the chronic stages of infection. However, existing methods for detecting sRNA require specialized equipment, making them unsuitable for field use. In this study, we have developed both a fluorescence-based and a lateral flow diagnostic test utilizing Cas10 technology for the detection of 7SL-sRNA from *Trypanosoma congolense* and *T. brucei*. The fluorescence assay detects 10 – 100 fM *T. congolense* 7SL-sRNA and 1 pM *T. brucei* 7SL-sRNA, and the lateral flow assay showed a limit of detection of 1 – 10 pM for both species. Either assay can effectively identify active infections in cattle, including during chronic phases (with positive signals observed up to the experimental end point, 63 days post infection). This also highlights the effective use of Cas10 for small RNA detection, paving the way for a cost-effective, user-friendly, and field-deployable diagnostic test for AT, while establishing Cas10 technology for the detection of small RNAs in general.

## Author summary

Animal Trypanosomosis (AT) is a parasitic disease that affects livestock, causing major economic losses and harm to animal health. Cheap effective diagnostics that can be easily used in the field would have a major impact on the disease burden by enabling better treatment practices. However, parasites causing the disease - trypanosomes - are often undetectable via conventional methods, especially during chronic disease stages. More

**Data availability statement:** All relevant data are within the manuscript and its Supporting Information files.

**Funding:** PCS, LJM and FG are funded through core support to the Roslin Institute by the United Kingdom Biotechnology and Biological Sciences Research Council (BBSRC) (BBS/E/RL/230002C); PCS is supported by a BBSRC Discovery Fellowship (BB/X009807/1); LJM and KRM were supported by a Wellcome Trust Collaborative Award (206815/Z/17/Z); MFW and SG are funded by a grant from the Biotechnology and Biological Sciences Research Council (Grant REF BB/T004789/1) and a European Research Council Advanced Grant (Grant REF 101018608). The funders had no role in study design, data collection and analysis, decision to publish, or preparation of the manuscript.

**Competing interests:** The authors have declared that no competing interests exist

accurate diagnostics are not suitable for use in the field and are therefore unavailable to farmers. This has led to the overuse of drug treatments that can harm animals and lead to the emergence of drug resistance. We previously discovered a trypanosome-specific small RNA, called 7SL-sRNA, that can be detected in the blood of infected cattle, even during the chronic stages of AT when parasites are not detectable via microscopy. However, detecting this RNA has required complex equipment, limiting its use in field settings. Building on this work, we have now developed a test for trypanosomosis that uses Cas10 technology to detect 7SL-sRNA via a simple lateral flow assay. This test is easy to use and accurately identifies active infections, including during the chronic phase of the disease. This is the first time Cas10 technology has been used to detect small RNAs using a lateral flow assay, offering a promising, cost-effective tool for diagnosing AT and potentially other neglected parasitic diseases.

## Introduction

African trypanosomes are extracellular hemoflagellate protozoan parasites that cause disease in both humans and animals [1]. The main species causing animal disease (Animal Trypanosomosis; AT) are *Trypanosoma congolense*, *T. brucei*, *T. vivax* and *T. b. evansi*. *T. congolense* and *T. brucei* are cyclically transmitted by tsetse flies (*Glossina* species), whereas *T. b. evansi* is transmitted mechanically by biting flies. T. vivax can be transmitted both cyclically and mechanically [2]. Cyclically-transmitted AT is mainly prevalent across sub-Saharan Africa, where it is restricted by tsetse fly distribution. However, mechanically-transmitted AT is prevalent in Latin America, North Africa and Asia. Subspecies of *T. brucei*, *T. b. gambiense* and *T. b. rhodesiense*, cause human disease (Human African Trypanosomiasis; HAT) across sub-Saharan Africa; *T. b. gambiense* (92% of cases) is restricted to West and Central Africa, and *T. b. rhodesiense* (8% of cases) causes infections in East and Southern Africa [3].

Significant efforts over the past decades have aided in substantially reducing the burden of HAT, with <1,000 cases reported in 2023, compared to >50,000 cases in the early 2000s [4]. This trend has led WHO to declare HAT as a target for elimination by 2030 [5]. HAT infection manifests in two stages: a haemolymphatic stage (first stage), after which parasites cross the blood-brain barrier causing the meningoencephalitic stage (second stage). Treatment depends on the sub-species and the disease stage, highlighting the importance of proper diagnosis.

In contrast to HAT, AT remains a significant threat to animal health across multiple continents. Whilst many different livestock (including goats, sheep and horses) are affected, the relative impact on cattle is the greatest, with >3 million deaths annually, and 90 million cattle at risk of disease in Africa alone [1]. The main methods of AT control are centred on chemotherapeutic treatment of active infection. However, continued use of drugs over many decades has led to frequent treatment failure, primarily attributed to parasite resistance. No new drugs have been approved in the past 60 years, and to ensure sustainable and reliable use of currently available drugs, diagnostics will be crucial to monitor disease outbreaks and prevalence, and to target drug treatment to infected animals.

Improved diagnostics are required for both HAT and AT, for the former to facilitate delivery of WHO objectives by 2030, and for the latter to reduce disease burden. Current recommended HAT diagnostics include visual identification of parasites via microscopy (in either blood, or cerebrospinal fluid [CSF} following lumbar puncture), serology-based assays (aimed at detecting antibodies against immunodominant antigens – developed into field-applicable Card Agglutination Tests), or analysis of CSF following lumbar puncture to detect either parasites or raised white blood cell counts for diagnosis of second stage HAT. PCR-based

approaches to detect parasite DNA have also been developed but are not currently part of the WHO-approved diagnostic process. Diagnosis of AT in the field primarily relies on clinical signs, although the symptoms, primarily anaemia, lymphadenopathy and ill thrift, are not AT-specific. Microscopy-based approaches can also be used, with the microhaematocrit centrifugation technique (MHCT) and the buffy coat technique (BCT) being sometimes utilised to concentrate parasites and improve sensitivity. Additionally, serology (ELISA) and PCR-based approaches have been developed, but there has been very poor uptake of the available ELISA kit (likely in part due to cost), and PCR is not field-applicable [6]. Additionally, antibody-based ELISAs suffer from the persistence of circulating antibody, meaning a positive result does not necessarily indicate an active infection. In general, these diagnostics approaches suffer from poor sensitivity and specificity, or cost and practicality limit their use in the field. As a result, improved diagnostics are a clear key requirement for disease management. Several recent studies have reported alternative approaches such as rapid diagnostic tests (RDTs – HAT only) [7], faecal sampling (AT, and this approach only detected *T. brucei*) [8], detection of biochemical biomarkers [9] and nanobody/monoclonal antibody sandwich technology [10]

Previously, we described a trypanosome-derived small RNA that is excreted/secreted at high levels into the extracellular environment during *in vitro* culture and infection of cattle, and therefore exhibits many characteristics that are desirable for a diagnostic test [11]. The 26-nucleotide 7SL-sRNA is derived from the long non-coding 7SL RNA (Fig 1A), and qPCR-based approaches enable robust detection in samples derived from both *in vitro* (culture supernatants) as well as *in vivo* (serum) infection settings, from the three species of African trypanosome responsible for the majority of AT and HAT cases (*T. congolense*, *T. vivax* and *T. brucei*). During experimental infections, 7SL-sRNA is detected even in periods when parasites are not detectable by microscopy, with fluctuating and intermittent parasitaemia a characteristic of trypanosome infections. The 7SL-sRNA sequence is species-specific, enabling the application of qPCR assays that discriminate between *T. congolense*, *T. vivax* and *T. brucei* with high specificity and sensitivity. Importantly, the assays do not cross-react with the commensal widely distributed and non-pathogenic cattle trypanosomatid, *Trypanosoma theileri* [12]. Furthermore, detection is indicative of active infection, as drug treatment of cattle rapidly led to complete ablation of 7SL-sRNA signal during experimental infections [11]. Therefore, 7SL-sRNA exhibits desirable traits for a highly sensitivity and specific diagnostic. However, studies to date have employed qPCR-based approaches for the detection of 7SL-sRNA, and as mentioned above, field-applicable tests are vital for routine diagnosis of HAT and AT, especially as the diseases mostly occur in settings where equipment such as qPCR machines are not readily available.

Recently, the use of CRISPR-Cas based approaches has enabled the development of highly sensitive and specific diagnostics. For example, using dCas9 for the detection of *Leishmania* spp. [13] and the Cas13a-based SHERLOCK approach for the species-specific detection of both gambiense-HAT and rhodesiense-HAT [14]. We recently demonstrated the potential for type III CRISPR-Cas systems for the development of highly sensitive RNA detection assays [15]. Specifically, we utilised the *Vibrio metoecus* Cmr (VmeCmr) effector, which includes a catalytic Cas10 protein with nucleotide cyclase activity, coupled with a NucC DNA nuclease effector. Cas10 is activated by target RNA binding, generating cyclic-triadenylate ($cA_3$) to stimulate NucC-mediated DNase activity in a highly specific manner (Fig 1B) [15].

In this study, we demonstrate detection of 7SL-sRNA from *in vitro* and *in vivo* samples using this Cas10 detection system. We show that 7SL-sRNA can be detected at femtomolar levels in a species-specific manner. In addition, we show that the Cas10 system can be harnessed for the development of lateral flow tests (LFTs). This approach leverages potential advantages of targeting an extra-cellular nucleic acid that can be detected at high levels in the

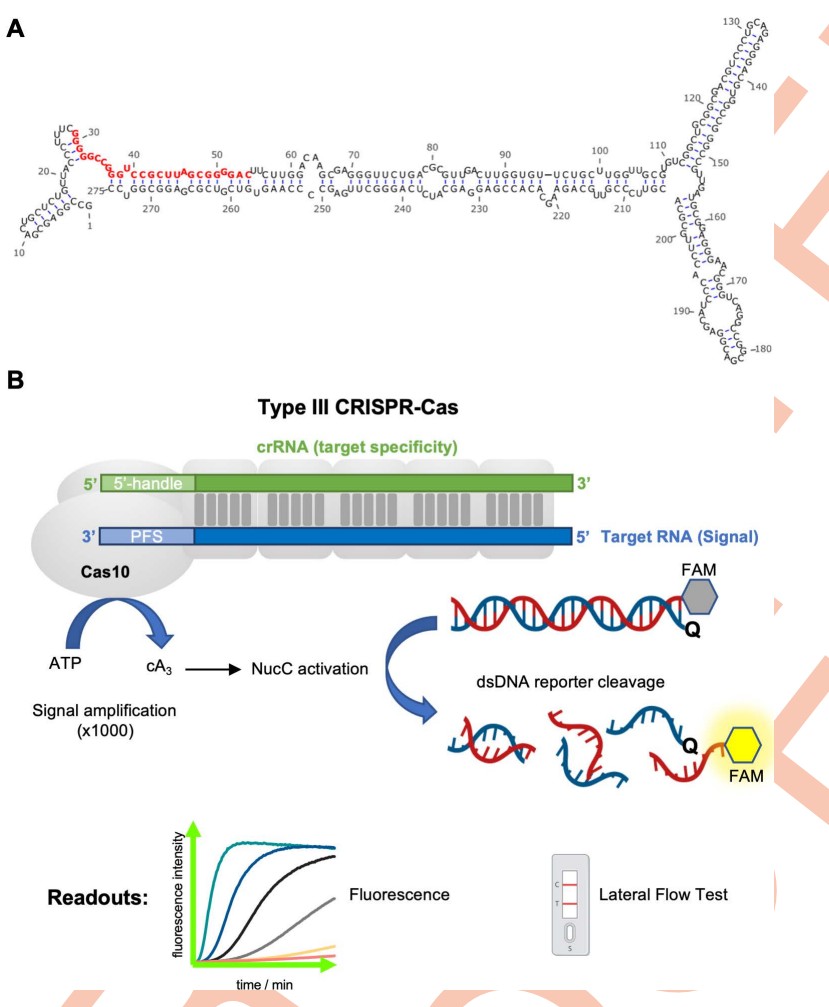

**Fig 1. A. Predicted secondary structure of the *T. congolense* full 7SL-RNA molecule.** The 7SL-sRNA fragment that serves as the diagnostic target is highlighted in red. B: Outline of type **III** CRISPR-Cas (Cmr)/NucC assay for the detection of RNA. Upon target RNA binding the cyclase (Palm) domain of the Cas10 subunit is activated to produce approximately 1,000 cA$_3$ (3'-5'-cyclic triAMP) molecules per target RNA. The cyclic oligoadenylate in turn activates the nuclease NucC to degrade the dsDNA reporter. Reporters labelled with a fluorophore:quencher pair as shown are monitored by following the development of fluorescence. The reporter can be adapted to enable detection by lateral flow test. Figure created with BioRender.com.

serum of infected animals, even during latent phases of infection. Additionally, 7SL-sRNA serves as a marker of active infection, enhancing its diagnostic value.

## Materials and methods

### Ethics statement

Animal experiments were previously carried out [8] at the Large Animal Research and Imaging Facility at the Roslin Institute, University of Edinburgh, under the auspices of United Kingdom Home Office Project License number PE854F3FC. Studies were approved by the Roslin Institute (University of Edinburgh) Animal Welfare and Ethical Review Board (study number L475). Care and maintenance of animals complied with University regulations and the Animals (Scientific Procedures) Act (1986; revised 2013) and with ARRIVE guidelines (https://arriveguidelines.org/).

### *In vivo* infections

Serum samples used in this study were derived from a previous *T. congolense* infection study carried out under vector-proof containment at the Large Animal Research and Imaging Facility at the Roslin Institute [8]. In this study, six male Holstein-Friesian cattle of post-weaning age (4-6 months) were inoculated with $1 \times 10^6$ trypanosomes (Savannah IL3000 strain) via the jugular vein. Infections were followed for 63 days and jugular blood samples taken every 2-3 days for PCV and parasitaemia measurements. Samples were analysed for the presence of parasites using the buffy coat technique, and parasitaemia was scored based on the number of trypanosomes observed in each preparation [16]. Surplus serum was stored at -80°C. From this cohort, serum samples derived from three calves (numbers 222, 230 & 267) were analysed for this study as outlined below. A total of 13 samples were analysed for each calf, including a -2 day (before infection) negative control.

### *In vitro* cell culture

*In vitro* culture supernatants were obtained from routinely cultured *T. congolense* and *T. brucei*. *T. congolense* (strain IL3000) was cultured in HMI-93 [16], supplemented with 20% goat serum (Gibco), and cells maintained at 34°C, 5% $CO_2$. *T. brucei* (strain Lister 427) was cultured in HMI-11 [17], supplemented with 10% foetal bovine serum (FBS; Gibco), and cells maintained at 37°C, 5% $CO_2$. For both species, cell densities were maintained between $2 \times 10^4$ and $2 \times 10^6$ cells/mL. Cell densities were monitored via haemocytometer using a phase-contrast microscope.

### RNA extractions

RNA was extracted from serum samples deriving from experimental cattle infections, or *in vitro* culture supernatants, using TRIzol LS reagent (Life technologies). All centrifugation steps were carried out at 4°C. A total of 250 μL serum was mixed with 750 μL TRIzol LS and incubation at room temperature for 10 min before addition of 200 μL chloroform. Samples were mixed vigorously and incubated for a further 10 min prior to centrifugation at 16,060 $\times g$ for 10 min. The aqueous phase (~500 μL) was transferred into a new sterile RNase-free Eppendorf and supplemented with 1 μL GlycoBlue coprecipitant (Life Technologies) and 500 μL isopropanol. Samples were mixed and incubated at room temperature for 10 min, followed by another centrifugation step at 16,060 $\times g$ for 15 min. The supernatant was poured off, the pellet washed with 700 μL ice-cold 75% ethanol, and the samples mixed before repeating the centrifugation step. Pellets were washed with 75% ethanol once more and centrifuged again before removing the supernatant by pipetting and air-drying the pellet in a laminar flow hood. Pellets were resuspended in 15 μL RNase-free water, and concentrations measured via a nano-drop spectrophotometer (Thermo Fisher Scientific).

### 7SL-sRNA detection by qPCR

qPCR analysis of 7SL-sRNA from serum samples or *in vitro* culture supernatants was carried out as described previously [11, 12]. Custom small RNA assays using a stem loop primer-probe approach were designed by Life Technologies, based on the species-specific 7SL-sRNA sequences (assay IDs: *T. brucei*: CTFVKNM; *T. congolense*: CTRWEM9). Extracted RNA samples were reverse transcribed using a commercial kit (Applied Biosciences; 4368814). A 15 μL reaction was set up with 1.5 μL 10× RT buffer, 0.3 μL Custom small RNA assay primer, 0.15 dNTPs (2 mM), 0.19 μL RNase inhibitor (Life Technologies; N8080119), 1.0 μL Multiscribe, 9.86 μL water and 2.0 μL RNA. The cycling conditions were as follows: 16°C for 30 min, 42°C for 30 min and 85°C for 5 min.

Detection of 7SL-sRNA was carried out via qPCR, using a commercial master mix (Thermo; 4304437). A 20 μL reaction was set up containing 10 μL 2× master mix, 1 μL Custom small RNA assay probe, 7.5 μL water and 1.5 μL cDNA obtained in the reverse transcription step. qPCR was carried out using a Rotor-gene Q (QIAgen) with the following cycling conditions: 50°C for 2 min, 95°C for 10 min, followed by 40 cycles of: 95°C for 15 seconds and 60°C for 1 minute. Probe detection was achieved at the 60°C step. Results from qPCR are expressed as the difference in Ct value to the negative control (Ct = 40) for ease of representation.

## Cloning

Enzymes were purchased from Thermo Scientific or New England Biolabs and used according to manufacturer's instructions. RNA and DNA oligonucleotides were obtained from Integrated DNA Technologies (Leuven, Belgium). All final constructs were verified by sequencing (Eurofins Genomics, Germany). The generation of expression constructs for *Vibrio metoecus nucC* (NucC), *Vibrio metoecus cmr1-6,* and *V. metoecus cas6f* as well as the methodology for incorporating specific targeting spacers into a *V. metoecus* minimal CRISPR array have been previously described [15]. CRISPR spacer sequences targeting *T. congolense* and *T. brucei* 7SL-sRNA are listed in Table 1. The first 21 nt of the spacer sequence are capable of base-pairing with the corresponding 7SL-sRNA.

## Protein production and purification

A detailed description for the purification of NucC and the crRNA-charged Cmr complex has been provided in reference [15]. Briefly, NucC was produced in *E. coli* C43 (DE3) [18] with an N-terminal, TEV protease-cleavable His$_8$-tag. The enzyme was purified by nickel-affinity chromatography, followed by cleavage of the His-tag and a final size exclusion chromatography step.

The crRNA-loaded Cmr complex was produced in *E. coli* BL21 Star (DE3) cells (Invitrogen) by co-production of the Cmr1-6 proteins and the mini-CRISPR array plus processing enzyme Casf. The enzyme was purified by nickel-affinity chromatography, followed by cleavage of the His-tag from Cmr3. The Cmr complex was further purified by size exclusion and heparin chromatography.

Protein concentrations were determined by UV spectroscopy using an extinction coefficient of 29,910 M$^{-1}$ cm$^{-1}$ for NucC (monomer) and 610,240 M$^{-1}$ cm$^{-1}$ for Cmr[crRNA]. Single-use aliquots of the enzymes were flash-frozen and stored at -70 °C.

**Table 1. Nucleic acid sequences.**

| Name | Sequence (5' → 3') |
| --- | --- |
| Tco 7SL-sRNA | GGGGGCCGGGUCCGCUUAGCGGGGAC (80.8% GC) |
| Tbr 7SL-sRNA | GGGGGCUGAUCCCGCUUAGCGGGGAC (73.1% GC) |
| Non-target RNA | AGGGUAUUAUUUGUUUGUUUCUUCUAAACUAUAAGCUAGUUCUG-GAGA |
| Tco spacer | CGCTAAGCGGACCCGGCCCCCgaaagggtaac (21 nt: Delta G -53.82 kcal/mol, Tm 76.2 °C) |
| Tbr spacer | CGCTAAGCGGGATCAGCCCCCgaaggttacag (21 nt: Delta G -49.34 kcal/mol, Tm 72.6 °C) |
| LFT dsDNA substrate | FAM-AGTGTTACATTATCCACCATGGCGAGCTTT-Biotin: AAAGCTCGCCATGGTGGATAATGTAACACT |
| Fluorescence dsDNA substrate | FAM-AGTGTTACATTATCCACCATGGCGAGCTTT: AAAGCTCGCCAT GGTGGATAATGTAACACT-Iowa Black |

### Fluorescent 7SL-sRNA detection assay

All assays were performed in duplicate on a FluoStar Omega plate reader (BMG Labtech) using fluorescence detection ($l_{ex/em}$ 485/520 nm) in black, non-binding half-area 96-well plates (Corning). Synthetic target RNAs are listed in Table 1; each Cmr-coupled nuclease assay contained 50 nM Cmr[crRNA] complex in 12.5 mM Tris-HCl, pH 8.0, 10 – 20 mM NaCl, 10 mM MgCl$_2$, 10% glycerol, 500 µM ATP, 125 nM FAM: Iowa Black double-stranded DNA substrate (Table 1) and varying concentrations of target or non-target RNA (Table 1) unless stated otherwise. NucC was added after 10 min at 37 °C to a final concentration of 250 nM. Fluorescence was measured throughout in 1 min intervals at 37 °C for up to 180 min. The signal between 2 and 9.5 min was used for baseline subtraction (Mars software, BMG Labtech). Statistical analyses were performed with Prism 10 (GraphPad). Samples giving rise to a 1.5-fold increase in fluorescence intensity relative to the negative control after 2 h reaction time were deemed positive.

### 7SL-sRNA detection by lateral flow test

The assay consisted of 50 nM Cmr[crRNA] complex in 12.5 mM Tris-HCl, pH 8.0, 10 – 20 mM NaCl, 10 mM MgCl$_2$, 10% glycerol, 500 µM ATP, 500 nM FAM-30mer-Biotin double-stranded DNA substrate (Table 1), 250 nM NucC and varying concentrations of target or non-target RNA unless stated otherwise. Reactions were incubated at 37 °C for 2 h unless stated otherwise. The reaction mixture was diluted ten-fold into HybriDetect Assay Buffer to 100 µl final volume, in which HybriDetect dipsticks (Milenia GenLine HybriDetect kit, Generon, UK) were developed for 3 min.

## Results

### Guide RNA design influences RNA detection sensitivity

Type III CRISPR systems can be programmed using a CRISPR RNA (guide RNA) to detect any desired RNA sequence. The guide RNA has an 8 nucleotide (nt) 5' handle (nt positions -8 to -1) that is derived from the CRISPR repeat and thus does not base pair with RNA targets. Target RNA in turn has a region known as the "protospacer flanking sequence" (PFS) opposite the 5' handle, and base pairing between them can be deleterious for Cas10 activation [19, 20]. For *V. metoecus* Cas10, we showed previously that a minimum of 5 nt PFS was required and that base-pairing between the PFS and the 5' handle was detrimental to cOA production [15].

The Cmr complex isolated from our overexpression system is a mixture of two main crRNA lengths, 40 nt and 34 nt, which can therefore base-pair with 32 nt or 26 nt of a target RNA (S1 Fig). We first investigated the effect of changing the lengths of the PFS and the base paired region of target RNAs on assay sensitivity. Using a synthetic 26 nt target RNA corresponding to the 7SL-sRNA sequence from *T. congolense* (Tco-7SL-sRNA), we tested Cmr complexes with different guide RNAs (Fig 2). The first, Tco_CA1, base paired with 21 nt of the target, leaving a 5 nt PFS. The second, Tco_CA2, could form 18 bp, leaving an 8 nt PFS. The limit of detection (LoD) was determined at 1 h reaction time for each complex using our NucC coupled fluorescence assay [15]. We observed a LoD of 100 fM for Tco_CA1 and 10 pM for Tco_CA2, demonstrating that target RNAs shorter that 26 nt can be detected sensitively, but that there is a detrimental effect on sensitivity with the smaller target RNA (Fig 2).

### V. metoecus Cmr can be programmed to detect T. congolense and T. brucei 7SL-sRNA

We designed a further mini-CRISPR array targeting *T. brucei* 7SL-sRNA (Tbr-7SL-sRNA) based on the Tco_CA1 guide RNA design, and the Cmr[7SL crRNA] complexes were

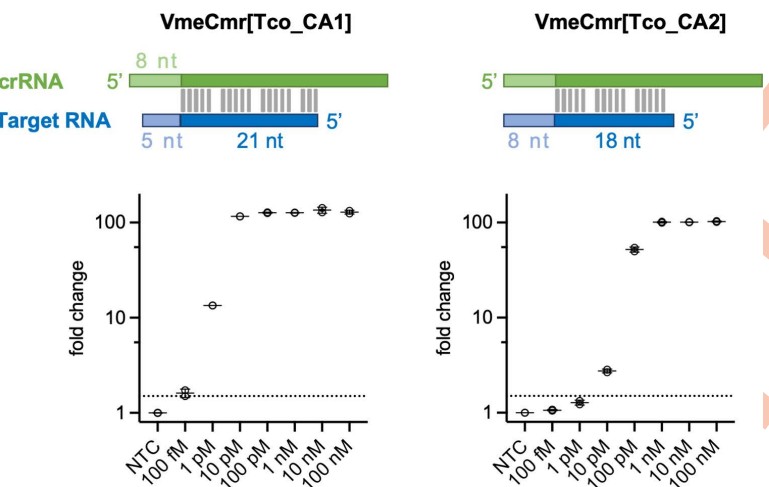

**Fig 2. Effect of crRNA design on assay sensitivity.** Extending the length for the PFS (light blue) at the cost of base-pairing decreases sensitivity 100-fold, from 100 fM (5 nt PFS) to 10 pM (8 nt PFS). The dotted line indicates 1.5-fold change in fluorescence intensity relative to non-target control which is set as the detection threshold. Mean of technical duplicates is shown.

expressed and purified. We tested both complexes against synthetic 7SL-sRNA targets to establish the limit of detection for both *T. congolense* and *T. brucei*. In our previous study with VmrCmr targeting the SARS-Cov-2 N gene a reaction time of 30 min was sufficient to obtain high sensitivity. Compared to the SARS-Cov-2 system, the sensitivity against trypanosomal 7SL-sRNA was lower. However, extending the reaction time increased the sensitivity and lowered the LoD 10-fold for both species. A 2 h reaction time tended to provide the best trade-off between experiment length, increased background and sensitivity and was therefore used for all further studies using the fluorescence assay (Fig 3).

We proceeded to test the LoD for 7SL-sRNA from both species, observing LoDs of 10 fM and 100 fM, respectively, for *T. congolense* and *T. brucei* (Fig 4). As each guide RNA recognised 21 nt of target RNA, the difference in detection sensitivity may be due to the differing GC content of the targets and/or their propensity to adopt stable secondary structures *in vitro*, which is known to influence detection sensitivity [15]. We also investigated cross-reactivity by assaying each of the Cmr complexes with synthetic 7SL-sRNA from the non-cognate species (Fig 4).

Cmr[Tco] showed minimal cross-reactivity with Tbr-7SL-sRNA (LoD 1 nM), but Cmr[Tbr] detected Tco-7SL-sRNA with an LoD of 1 pM, 1 order of magnitude higher than the LoD of 100 fM observed for the cognate Tbr-7SL-sRNA species. Overall, these data highlight that type III CRISPR systems can detect short target RNAs with sensitivity in the femtomolar range, without any pre-amplification step. LoDs must still be determined empirically, as the large variations observed cannot necessarily be predicted and may often involve target RNA secondary structure formation.

## Detection of Trypanosomal 7SL-sRNA from *in vitro* supernatant RNA extracts

To investigate more complex samples than synthetic RNA, *in vitro* time courses were initiated with either *T. congolense* or *T. brucei* at seeding densities of $5 \times 10^4$ cells/mL and $5 \times 10^5$ cells/mL for *T. congolense* and *T. brucei*, respectively. Cell culture supernatants were periodically sampled and RNA isolated from 0 to 96 h culture. 7SL-sRNA was analysed by RT-qPCR as

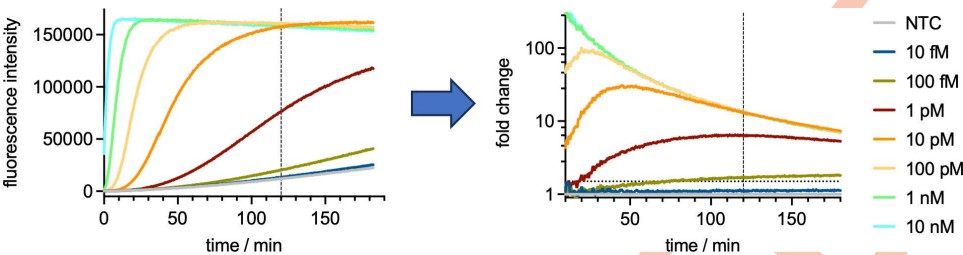

**Fig 3. VmeCmr/NucC assay time for LoD determination.** The development of fluorescence intensity in response to target RNA is measured over time. The fluorescence intensity is then expressed relative to the non-target (negative) control sample. Any change in intensity >1.5-fold (horizontal dotted line) higher than the negative control is taken as positive for the presence of target RNA. Vertical line indicates 120 min as the optimal read time based on experiment length, background signal and sensitivity. The example shown is for the *T. brucei* 7SL-sRNA-targeting Cmr using synthetic Tbr-7SL-sRNA as the target.

well as by Cmr/NucC assay (Fig 5). For both species, the Cmr/NucC fluorescence assays

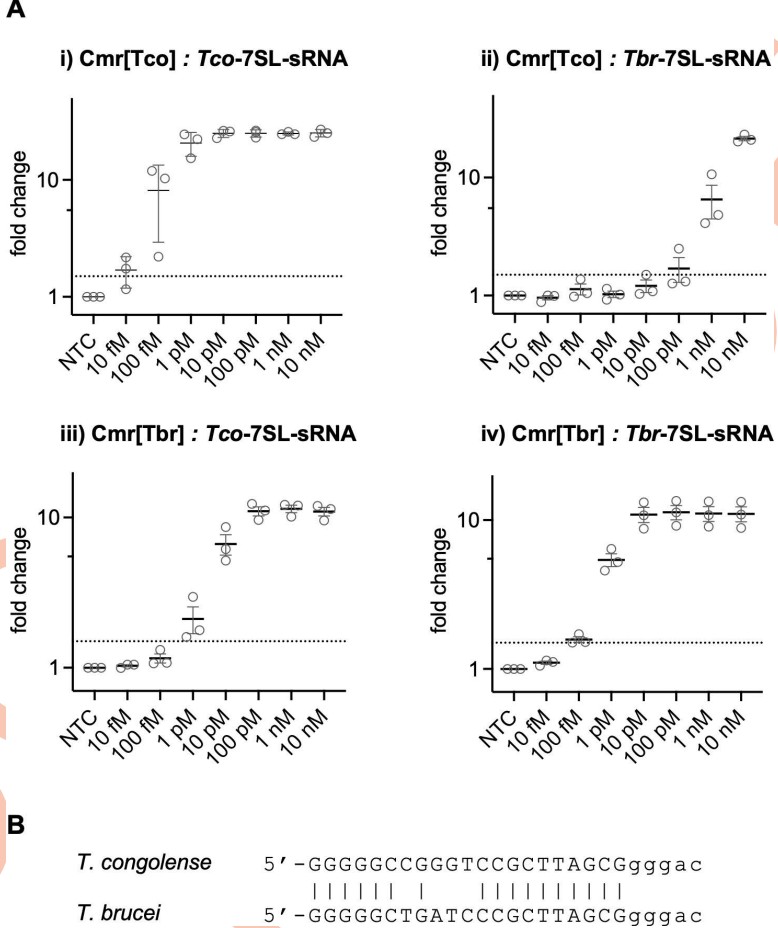

**Fig 4. Limit of detection for *T. congolense* and *T. brucei* and discrimination against related species.** A: Synthetic 7SL-sRNA from either species was tested with Cmr[Tco] and Cmr[Tbr] complexes in the Cmr/NucC fluorescent assay. The results give the change in fluorescence intensity relative to the non-target control. The dotted line indicates a 1.5-fold change in fluorescence and demarcates the detection threshold. The mean values with SD from 3 independent experiments are shown. NTC: non-target control. B: Pairwise alignment of the two 7SL-sRNA species used in this study.

performed as well as the previously developed RT-qPCR [11], with all test results in agreement. Notably, when we performed the fluorescence assay with Cmr[Tbr] on the non-cognate *T. congolense* samples, no false positives were detected despite the low LoD of Cmr[Tbr] for Tco-7SL-sRNA (S2A Fig). Furthermore, none of the *T. brucei* samples tested positive when Cmr[Tco] was used. The successful discrimination between cognate and non-cognate target by Cmr[Tbr] in this experiment is due to the low concentration of *T. congolense* target RNA present as judged by comparison to the positive control using synthetic 7SL-sRNA (S2B Fig).

## Detection of 7SL-sRNA by Lateral Flow Test

The Cmr/NucC assay was adapted for a lateral flow format by using a dsDNA reporter with a FAM label on one end and biotin on the other (Fig 6A). In the lateral flow test (LFT), gold nanoparticles (GNPs) conjugated to FAM antibodies migrate up a membrane by capillary forces and interact with two different sections. In the first section the control (C) line is coated with streptavidin to capture a biotinylated reporter; if the reporter is intact, the FAM label is bound by the anti-FAM antibody:GNP conjugate. In the second section, the test (T) line is coated with a secondary antibody that binds the anti-FAM antibody on the GNP. When the dsDNA reporter is degraded, no GNPs bind to the C line but are captured at the T line instead. Bound GNPs appear as a dark red line on the membrane. Using this method, we detected 10 pM of cognate synthetic 7SL-sRNA for both the *T. congolense*- and the *T. brucei*-targeting Cmr complexes (S3 Fig). Fig 6C and 6D shows the LFT results for the cell culture-derived samples described earlier. While the LFT is less sensitive than the fluorescence assay, the last four time points of each culture gave clear positive results.

## Detection of 7SL-sRNA from serum of infected cattle

Previously, we demonstrated that 7SL-sRNA detection by RT-qPCR is more reliable and sensitive than microscopy approaches for detecting trypanosomosis in experimentally infected cattle [11, 12]. To determine how the Cas10 assays performed in comparison, we tested serum samples previously collected from three calves over a long-term (63 days) infection with *T. congolense* [8]. After the initial phase (2-3 weeks), infection in cattle typically becomes chronic, characterised by long periods of very low or absent parasitaemia that is challenging to detect via conventional methods. RNA was extracted from serum samples and tested for detection of 7SL-sRNA by RT-qPCR, Cmr/NucC fluorescence assays and LFTs. The presence of *T. congolense* was only detected from day six post-infection by microscopy. Thereafter, the

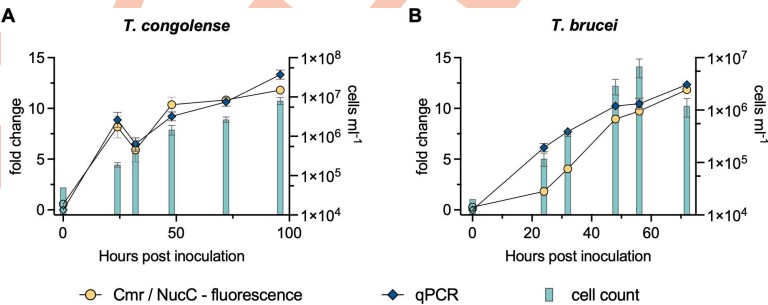

**Fig 5. 7SL-sRNA detection from *in vitro* *T. congolense* (A) and *T. brucei* (B) cell culture supernatants.** Fold change for Cmr/NucC fluorescence and RT-qPCR assays are shown on the left *y*-axis. The parasite cell count is shown on the right *y*-axis. Filled symbols indicate a positive test (above threshold), whereas open symbols indicate a negative test result for the Cmr/ NucC and qPCR assays. The mean and SD from triplicates are shown.

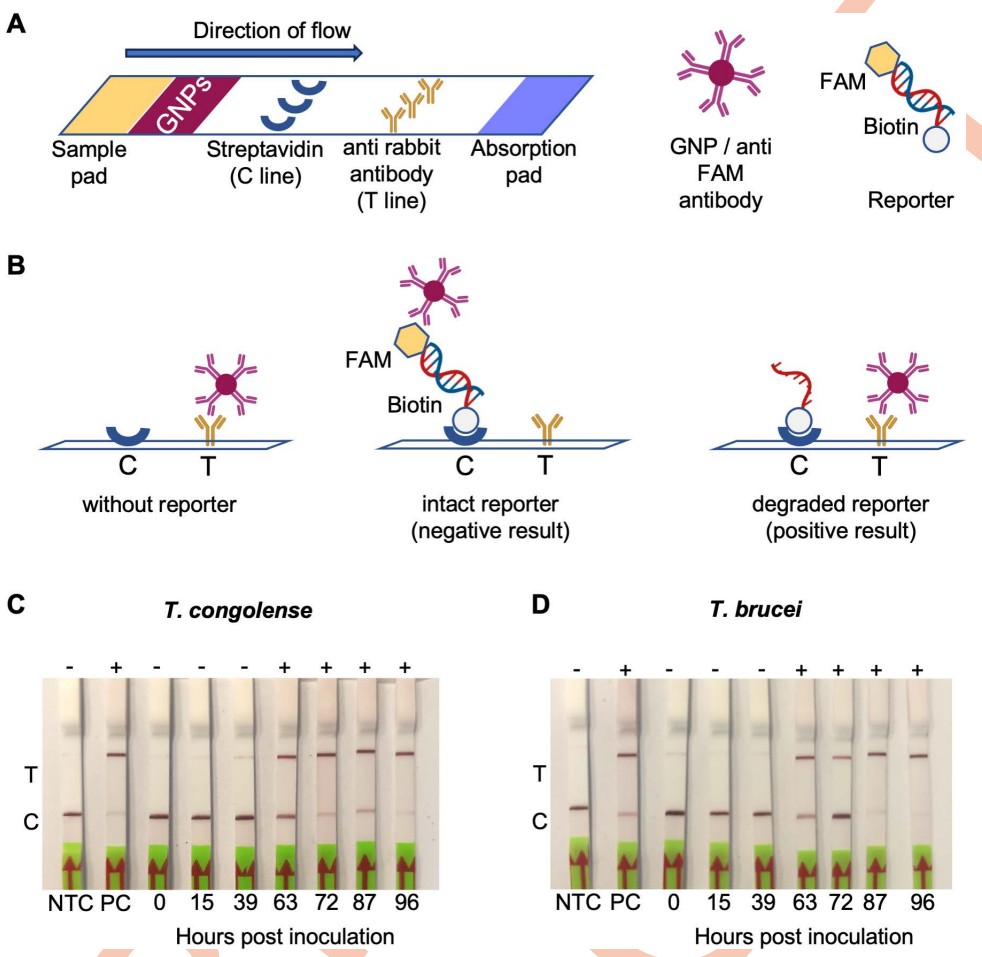

**Fig 6. Detection of Tbr- and Tco-7SL-sRNA using the Cmr/NucC assay with LFT read-out.** A: Schematic representation of the LFT components. The lateral flow strip contains an area for sample application that is followed by deposited gold nanoparticles (GNPs) conjugated to anti FAM antibodies giving a deep purple colour. The sample migrates towards the adsorption pad with capillary force taking the GNPs along. First a line of immobilised streptavidin (control or C line) is traversed, subsequently a line of anti-rabbit antibodies (test or T line). The dsDNA reporter for the LFT is labelled with FAM, the ligand for the GNP conjugate, on one end and with biotin for streptavidin binding at the other end. B: Mechanism of varying test results for the LFT. In the absence of reporter, the GNP conjugate binds to the antibody at the T line. Both the intact reporter and the biotin-labelled degraded reporter after cA$_3$-activated NucC cleavage bind to streptavidin at the C line. However, the GNP conjugate can only bind to the C line when the reporter is intact and labelled with FAM. Any unbound GNP conjugates will then bind to the T line. The reporter concentration must be chosen carefully to obtain meaningful results. C and D: LFT tests were performed on RNA extracted from cultures inoculated with *T. congolense* (C) or *T. brucei* (D). The RNA extracts were the same ones used for the assays shown in Fig 5A and 5B, respectively. NTC: non-target control; PC: positive control, 10 pM cognate synthetic 7SL sRNA. The results of each LFT is indicated above the strip with "-" for a negative test result (no 7SL sRNA detected), and with "+" for a positive test result (7SL sRNA detected). Figure created with BioRender.com.

results varied between animals with parasites only detected in nine, six and four out of ten samples for animal 222, 230 and 267, respectively, indicating low sensitivity and a high rate of false negatives (Fig 7). In contrast, the presence of *T. congolense* was detected throughout the infection time course for all three animals via RT-qPCR. Using the Cmr/NucC fluorescence assay, we detected the parasite RNA in two of three animals at the earliest time point post-infection and in all three animals thereafter for the remainder of the time course. Similar to parasite counts, 7SL-sRNA was detected from day six post infection using LFT. However, the

7SL-sRNA LFT outperformed microscopy, with only four false negative results in all samples after six days post-infection. While it should be noted that the concentration of Tco-7SL-sRNA in some of these samples was in the 10 – 100 pM range, which brought them above the LoD for the non-cognate Cmr[Tbr] (S4 Fig), the results indicate that the 7SL-sRNA LFT is sensitive and accurate and can detect infection during the chronic infection phase when parasitaemia is undetectable by microscopy.

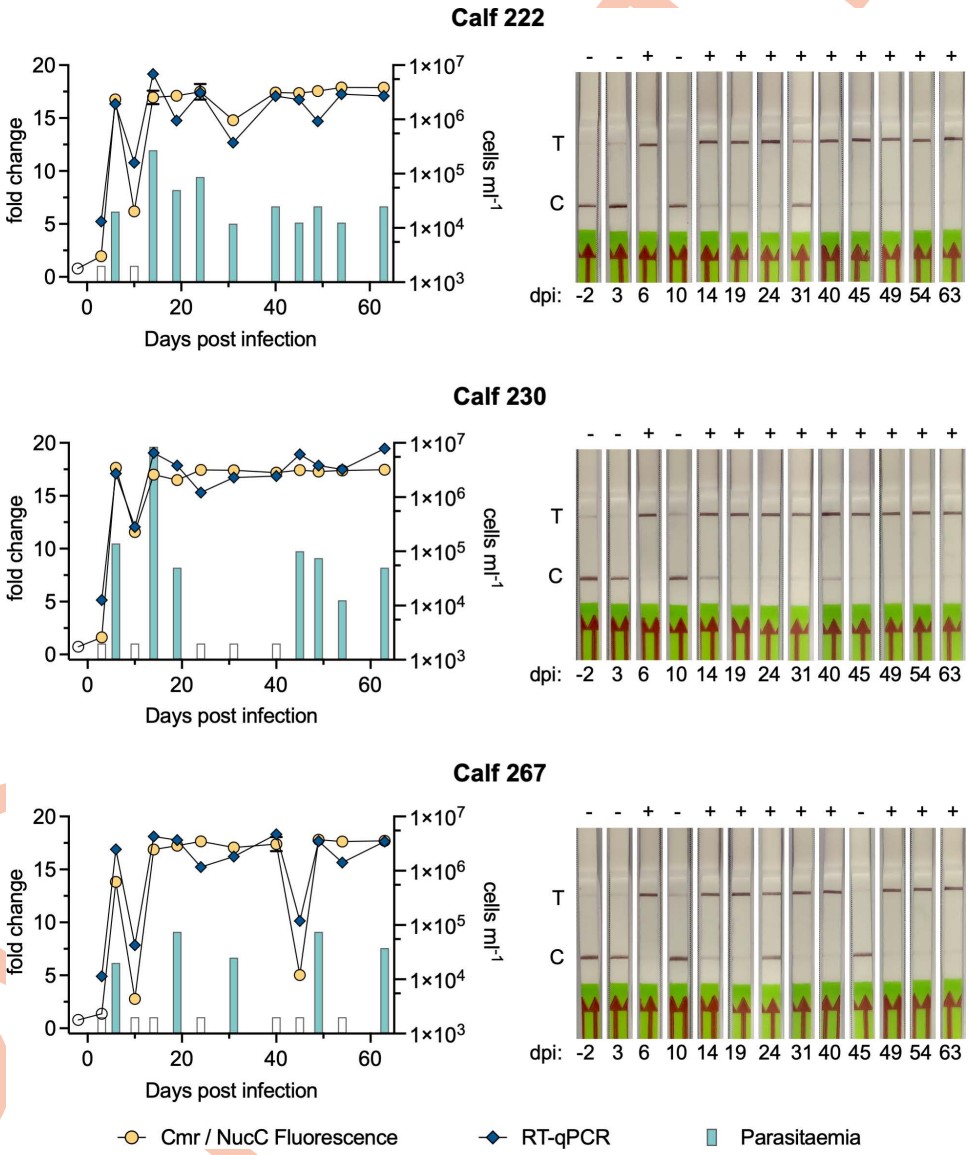

**Fig 7. Cas10-based detection of 7SL-sRNA during *in vivo* infection.** RNA was isolated from serum samples derived from three calves experimentally challenged with *T. congolense* for 63 days. The graphs show the results for each animal from Cmr/NucC fluorescence (circles) and RT-qPCR (diamonds) assays (left *y*-axis expressed as change in signal relative to negative control), and microscopically determined parasitaemia (bars, right *y*-axis). Filled symbols indicate a positive test (above threshold), whereas open symbols indicate a negative test result. The results from the LFT with Cmr[Tco] for the same samples are shown on the right with the test result indicated above each strip (+ for positive, - for negative). dpi: days post infection; T: test line; C: control line.

## Discussion

Field applicable diagnostics centred on the detection of RNA are scarce, and indeed most methodology is based on the detection of amplified, reverse transcribed RNA (e.g., sequencing, RT-qPCR) (reviewed in [21]). However, the detection of pathogen-specific RNA remains of interest, as circulating levels are normally multitudes higher than DNA (genome copy number) during active infections. Small RNAs (in particular miRNAs) have garnered increased interest in recent years as biomarkers for a variety of diseases [22], but are especially intriguing as diagnostic biomarkers for pathogens, because a positive result relies purely on RNA detection, not relative abundance. We recently characterised a novel small RNA, termed 7SL-sRNA, excreted at high levels by African trypanosomes - extracellular protozoan blood-borne parasites [11, 12]. African trypanosomes cause both human and livestock infection, particularly in low- and middle-income countries, and sensitive and specific field-applicable diagnostics are urgently required for accurate screening of both diseases; to facilitate elimination of HAT, and to control AT.

In this study, we have developed technology towards a field-applicable assay to detect the 7SL-sRNA, using both *in vitro* and *in vivo* sample analysis. We outline both fluorescence-based and LFT-based methodologies for 7SL-sRNA using a type III-B CRISPR system from *V. metoecus* that activates an enzymatic Cas10 subunit. The Cas10 subunit lacks an HD nuclease domain, but instead generates cOAs that stimulate NucC-mediated DNase activity. This system was previously characterised and incorporated into a highly sensitive and specific assay for the detection of SARS-CoV-2 RNA [15]. Importantly, this assay requires no extrinsic RNA amplification [15] and is isothermic. Using this system, we were able to detect synthetic trypanosomal 7SL-sRNA at femtomolar concentrations, using gRNAs that pair with 21 nt of the 26 nt sRNA target (leaving a 5 nt PFS). Notably, extending the PFS to 8 nt had a detrimental effect on sensitivity (100-fold), highlighting a trade-off between sensitivity and specificity regarding the engineering of Cmr for small RNA detection.

Using a fluorescence-based assay incorporating the VmeCmr and NucC nuclease, detection of 7SL-sRNA from both *T. brucei* and *T. congolense* was specific and sensitive, with LoDs of 10 fM and 100 fM, for *T. congolense* and *T. brucei*, respectively. Given these results, we next developed an LFT-based approach to detect the trypanosome 7SL-sRNA, using the VmeCmr, NucC nuclease and a dsDNA reporter linked to both biotin and FAM. Under these conditions, we detected 10 pM of cognate synthetic 7SLs-RNA, as well as 7SL-sRNA derived from *in vivo* culture supernatants and *in vivo*-derived serum.

CRISPR-based diagnostics have previously been reported for human protozoan diseases including *Plasmodium falciparum* (Malaria) and *Trypanosoma brucei* (Human African Trypanosomiasis), using Cas12a [23] and Cas13a [14], respectively. These assays target genes within the parasites and require reverse transcriptase-recombinase polymerase amplification.

To the best of our knowledge, there is only one other example where a Cas10-based CRISPR-Cas system has been used to detect short RNA. The type III-A complex from *Lactobacillus delbrueckii* LdCsm has been used to detect miR-155a, a breast cancer-related miRNA species, with a limit of detection of 500 pM [24]. The LdCsm system makes use of its integral, target RNA-dependent DNase activity, an activity that is absent in the VmeCmr complex described here; however, this also obliterates the advantages of cOA-mediated signal amplification and does explain, at least in part, the $10^4$-fold lower sensitivity of LdCsm compared to the VmeCmr/NucC system. It is likely that the 20 – 24 nt long miRNAs form the shortest RNA pool that can be accessed by type III CRISPR-Cas complexes and pushes the limits of what is feasible with Cas10 enzymes. The target-defining crRNAs of the type III family tend to be between 30 – 40 nt long, and lose cOA synthesis and DNase activity (if present) when targets shorter than 17 nt complementarity (plus the required 5 nt non-base pairing PFS) are

presented [20,24,25]. We were able to overcome the lower levels of cOA synthesis, resulting from the short target sequence, by simply extending the reaction time to maintain physiologically useful fM (or attomoles per reaction) detection limits. This brings the VmeCmr/NucC detection system in line with the more widely studied and employed Cas13 platforms or coupled Cas13/Cas14 that can detect miR species in low fM concentrations or even lower if pre-amplification steps or microfluidic target-ultralocalisation is employed [26–29].

There are notable advantages for utilisation of the system developed here compared to RT-qPCR detection of the 7SL-sRNA. In particular, no amplification method, such as Recombinase Polymerase Amplification (RPA), is required for the VmeCmr system, shortening the assay time and reducing assay complexity. Analysing excreted/secreted extracellular sRNAs means that a parasite cell does not need to be present in the sample, and as discussed earlier the numbers of sRNA targets will exceed the number of parasite cells manyfold; both features substantially enhance sensitivity. Additionally, the widespread familiarity of people with LFT assays (e.g., COVID-19 LFTs worldwide, and malaria rapid diagnostic tests are widely used across sub-Saharan Africa) means that this format is ideally suited to ease of use and interpretation by users.

Whilst this study demonstrates the translation of a small RNA assay towards a field-applicable platform, there are limitations that require further optimisation. Firstly, there is a loss in both sensitivity and specificity associated with the LFT format (e.g., false negative results and cross-reaction between species-specific assays, albeit at log-fold lower sensitivity), which could be addressed via further optimisation of the LFT methodology, including incubation time, reagent concentrations, control/test line concentrations and buffering conditions. Secondly, our approach still utilises RNA isolated via phenol:chloroform (TRIzol) extractions, which will not be feasible in point-of-care settings. Optimisation of simple RNA extraction buffers, for example, using acid pH [30], or by exploiting filter membrane-based extraction approaches [31], would enable widespread use of RNA-based LFTs. However, even with these limitations, it should be stressed that the *in vivo* samples assayed in this study included those from infection timepoints that represented the chronic infection low-parasitaemia status that is common in the field, and which is one of the biggest challenges for effective diagnostics. Therefore, the fact that the LFT assay was able to detect infection in this challenging phase of the disease is promising that the 7SL-sRNA target and LFT detection system may provide a route to a pen-side diagnostic with utility in the field.

Given the success of this assay in an experimental *in vivo* setting, further work will be required to ascertain the sensitivity and specificity of the LFT in a field setting, where livestock may often have co-infections with other pathogens (or even multiple trypanosome species). We previously showed that 7SL-sRNA qPCR assays do not cross-react with a commensal trypanosome, *T. theileri* [12], and similarly, we would not expect a Tbr- or Tco-7SL-sRNA-specific LFT to cross-react with this widespread species.

The successful application of the VmeCmr system to detection of small RNAs potentially extends to diagnostic uses in other contexts, including detection of orthologous 7SL-sRNAs or from other kinetoplastid parasites of economic importance. The system could also be adapted for the detection of sRNAs produced by other pathogens. There is also significant interest in the detection of microRNAs as markers of various disease states. Application of the VmeCmr system therefore expands the available tools for the detection of small RNAs. The lack of a need for direct amplification of the target nucleic acid and the development of LFT readout particularly lends this approach to field deployment and bedside tests.

In summary, we have adapted the detection of parasite-specific small RNAs towards a field-applicable LFT that is able to detect the presence of trypanosomes even during chronic infection stages, when parasites are typically not visible via microscopy-based approaches. The

data presented here show that RNA based diagnostics for neglected tropical diseases remain a significant possibility and bring us a step closer to field-applicable diagnostic tools to combat these diseases.

## Supporting information

**S1 Fig. Isolation of crRNA from Cmr complexes. RNA was isolated from the purified ribonucleoprotein complexes by phenol:chloroform extraction, 5'-end-labeled with [$^{32}$P] and analysed by denaturing PAGE.** Lane 1: Tco_CA1 crRNA; lane 2: Tco_CA2 crRNA; lanes 3 and 4: crRNA not discussed in the manuscript; lane 5: synthetic RNA oligonucleotide (44 nt).
(TIF)

**S2 Fig. 7SL-sRNA detection from in vitro T. congolense (i) and T. brucei (ii) cell culture supernatants. For each time time, RNA was isolated in triplicate and subjected to fluorescence-based Cmr/ NucC assay.** A: Increase in fluorescence after 2 h relative to non-target control (NTC). Samples were analysed with the cognate and non-cognate Cmr complex as indicated. Each replicate is shown with SD. B: Signal curves for the fluorescent assay using the cognate Cmr complex alongside selected synthetic standards to estimate 7SL-sRNA concentration in cell culture samples. Only the mean is shown for clarity. The majority of *T. congolense*-inoculated samples are above 1 pM and go up to around 10 pM. Surprisingly, these samples did not test positive with the Cmr[Tbr] complex (limit of detection for *Tco* 7SL-sRNA 1 pM, Fig. 4). This could suggest that specificity is increased rather than decreased in more complex samples. All of the *T. brucei*-inoculated samples contain 7SL-sRNA below the limit of detection of 1 nM for the non-cognate Cmr[Tco] complex (Fig. 4).
(TIF)

**S3 Fig. Establishing LoDs for T. congolense and T. brucei synthetic 7SL-sRNA using the Cmr/ NucC assay with LFT read-out. LFT limit of detection was tested using synthetic Tco-7SL-sRNA with Cmr[Tco] and synthetic Tbr-7SL-sRNA with Cmr[Tbr].** The concentration of the synthetic RNA is indicated above the strips. NTC: non-target control; T: test line; C: control line.
(TIF)

**S4 Fig. Cmr/NucC fluorescence assay for *in vivo* infection samples. The change in fluorescence signal relative to the non-target control using cognate Cmr[Tco] and non-cognate Cmr[Tbr] is shown in A.** Due to the high concentration of Tco7SL-sRNA in these samples (up to between 10 – 100 pM as judged by comparison to synthetic standards as shown in B), a high number of samples tested positive with the non-cognate Cmr[Tbr] complex. The reaction time was 2 h in A, and 2 and 20 min in B as indicated above the graph. Mean values plus SEM are shown except for the samples in B where only the mean fold change is plotted.
(TIF)

**S1 Data.**
(XLXS)

## Author contributions

**Conceptualization:** Sabine Grüschow, Pieter C Steketee, Liam J Morrison, Malcolm F White, Finn Grey.

**Data curation:** Sabine Grüschow, Pieter C Steketee.

**Formal analysis:** Sabine Grüschow, Pieter C Steketee.

**Funding acquisition:** Liam J Morrison, Malcolm F White, Finn Grey.

**Investigation:** Edith Paxton.

**Methodology:** Sabine Grüschow, Pieter C Steketee, Liam J Morrison.

**Resources:** Keith R Matthews.

**Supervision:** Liam J Morrison, Malcolm F White, Finn Grey.

**Validation:** Pieter C Steketee.

**Writing – original draft:** Sabine Grüschow, Pieter C Steketee, Liam J Morrison, Malcolm F White, Finn Grey.

**Writing – review & editing:** Sabine Grüschow, Pieter C Steketee, Keith R Matthews, Liam J Morrison, Malcolm F White, Finn Grey.

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
