## [Decision Letter · Decision Letter 0]

18 Oct 2024

Dear Career Track Fellow Grey,

Thank you very much for submitting your manuscript "Cas10 based 7SL-sRNA diagnostic for the detection of active trypanosomiasis" for consideration at PLOS Neglected Tropical Diseases. As with all papers reviewed by the journal, your manuscript was reviewed by members of the editorial board and by several independent reviewers. In light of the reviews (below this email), we would like to invite the resubmission of a significantly-revised version that takes into account the reviewers' comments. 

The manuscript is overall appreciated by the independent experts and editorial board members. Please address the comments and provide the data that are missing, especially with regards to determining the analytical sensitivity and specificity. Please also carefully report the replicates and standard deviations. This will require a major revision of the manuscript and its figures.

We cannot make any decision about publication until we have seen the revised manuscript and your response to the reviewers' comments. Your revised manuscript is also likely to be sent to reviewers for further evaluation.

Sincerely,

Guy Caljon

Academic Editor

Laura-Isobel McCall

Section Editor

The manuscript is overall appreciated by the independent experts and editorial board members. Please address the comments and provide the data that are missing, especially with regards to determining the analytical sensitivity and specificity. Please also carefully report the replicates and standard deviations. This will require a major revision of the manuscript and its figures.

Reviewer's Responses to Questions

**Key Review Criteria Required for Acceptance?**

**Methods**

-Are the objectives of the study clearly articulated with a clear testable hypothesis stated?

-Is the study design appropriate to address the stated objectives?

-Is the population clearly described and appropriate for the hypothesis being tested?

-Is the sample size sufficient to ensure adequate power to address the hypothesis being tested?

-Were correct statistical analysis used to support conclusions?

-Are there concerns about ethical or regulatory requirements being met?

Reviewer #1: Yes

Yes

Yes

Yes

Yes

No

Reviewer #2: (No Response)

Reviewer #3: Methods are clearly stated and appropriate.

**Results**

-Does the analysis presented match the analysis plan?

-Are the results clearly and completely presented?

-Are the figures (Tables, Images) of sufficient quality for clarity?

Reviewer #1: Yes

Yes

Yes, except for figures 2, 4, 5, and Fig S4

Reviewer #2: (No Response)

Reviewer #3: Overall the results are presented well.

**Conclusions**

-Are the conclusions supported by the data presented?

-Are the limitations of analysis clearly described?

-Do the authors discuss how these data can be helpful to advance our understanding of the topic under study?

-Is public health relevance addressed?

Reviewer #1: Yes

Yes

Yes

Yes, it is stated in the manuscript that the technology may be employed in detection of other diseases of public health importance.

Reviewer #2: (No Response)

Reviewer #3: Conclusions are overall supported, limitations are discussed.

**Editorial and Data Presentation Modifications?**

Reviewer #1: Minor Revision

Reviewer #2: (No Response)

Reviewer #3: Minor points:

1) Line 57 add (T) after Trypanosoma

2) Line 59 – one bracket

3) Line 134 – 135 Cas13a is RNA detecting not DNA: see - Abudayyeh, O. O., et al. RNA targeting with CRISPR-Cas13; Nature 2017 Vol. 550 Issue 7675

**Summary and General Comments**

Reviewer #1: Reviewers’ summary 

Animal trypanosomiasis remains one of the burdens affecting livestock industry in sub-Saharan Africa. While drugs for treatment of the disease are mostly potent, the existing challenge is diagnosis of infection. Current tests are either less sensitive, less specific or not be applicable in field situation. For this reason, veterinary practitioners often give blanket therapy. Gruschow et al developed a Cas10-based lateral flow assay (LFA) prototype for diagnosis of animal trypanosomiasis with potential for field application. Biomarker of detected by the assay is 7SL-sRNA of trypanosome present in culture supernatant and sera of infected animals. The work was built on their initial detection of the 7SL-sRNA of trypanosome released in clinical specimen by qPCR. However, due to the high demand for a field adaptable test, the authors went on to translate the laboratory-based qPCR into a LFA prototype. So far, the authors demonstrated that their novel LFA prototype test is capable of detecting 7SL-sRNA of trypanosomes in the invitro culture supernatant as well as sera of experimentally infected cattle. The development of field applicable assay for animal trypanosomiasis is a great step in the right direction that will boost efforts towards elimination of trypanosomiasis.

Strength of the study:

-The authors build on their previous work.

-Novelty of the work, Cas10 and 7SL-sRNA, is appreciated.

-Illustrative diagrams improved clarity of the work.

-The study was meticulously conducted and logically reported. 

-The prototype evaluated alongside three other tests (RT-qPCR, fluorescence and microscopy). 

-The authors exhibited honesty by citing key limitations of their study.

Comments: 

1. The use of words “trypanosomiasis” in the title text (line 2 & 3) and “trypanosomosis” in the body of the text (line 20, line 38, line 57 etc…) may be confusing to non-specialist readers. I would advise the authors to adopt any one of the two words for the entire text as appropriate. 

2. Line 117. Please revise the sentence to bring 7SL-sRNA “inside” the sentence.

3. Line 126. Define abbreviation cA3 in Fig 1 legend. 

4. Line 205. The word “each” carries a capital “E”. Should be changed to a small letter “e”.

5. Line 231. The sentence: “…transferred to a new sterile tube…”. The word “to” should be replaced with “into”.

6. Line 271. The sentence reading “The second, Tco_CA2 could…”. Please, insert a comma immediately after CA2.

7. Line 215. To ensure a logical flow of information in the Materials and Methods, I would advise that authors to describe RNA extraction first followed by qPCR and then end with LFA. 

8. Line 276. Increase the resolution of Figures 2, 4, 5, and Fig S4. 

9. Line 293 and 294 (Fig 3). For purpose of clarity, the authors should distinctly label the two graphs. Graph on the left side could be assigned letter “A”, and right-hand “B”. Could the author provide explanation why the curves for 100 pM to 10 nM are missing from the graph of “fold change vs time” (right hand-side)?

10. Line 297. The “non-target control” should be specified.

11. Line 309-Line 310 (Figure 4). The graphs should be made distinct from the aligned sequences. For example, assign the graphs as “A” with sub-levels (i-iv) and the aligned sequences as B. Thereafter, in each case, make a proper description of the figures in the legend.

12. Line 360. It reads “Fig 6B and C shows the LFT results…”. It should read “Fig 6C and D”, rather, not B. Fig 6B are illustrations meant for interpretation of the assay's results.

13. Line 377. C, D should be written as C and D.

14. Line 386. Application of microscopy for examination of sample should be mentioned/brought earlier in the Materials and Methods or, at least, be referenced. I only encountered it in the results section.

15. Line 479 - 484. I am not in agreement with the statement made by the authors that DNA and conventional RNA detection techniques require extraction of these biomarkers from intact trypanosomes yet the “new” 7SL-sRNA biomarker is released in the sample and, therefore, can be detected by their newly developed tool even in the absence of Trypanosoma cells. Unless experimentally proven, the fact that they had to extract RNA prior to detecting 7SL-sRNA may mean this biomarker is located intracellularly and only released in the milieu by decomposing cells. In my opinion DNA or any other forms of RNA are often released by decomposing Trypanosoma cells. The statement indicating that 7SL-sRNA can be detected without the need for intact parasite may be subjective. You may make this clear by answering the following questions: i) How long does the 7SL-sRNA biomarker remains detectable after clearing trypanosomes by drug treatment or after self-cure? ii) What mechanism does trypanosome use to excrete/secrete 7SL-sRNA? 

16. Line 509. What are the levels of sequence similarity between the 7SL-sRNA of T. brucei or T. congolense and other pathogenic species of animal trypanosomes when compared to T. theileri? Could the sequence divergence offer explanation why the tests did not cross-react with T. theileri?

Reviewer #2: This manuscript presents a novel and promising CRISPR-Cas10 based assay for the detection of trypanosome-specific 7SL-sRNA that could be applied to the diagnostic of T. congolense and T. brucei infections. 

Major comments

• Along the entire manuscript, human African trypanosomiases and animal African trypanosomiases are sometimes mixed, which is confusing. Please, carefully distinguish the human and animal diseases, homogenize the terminology, and clearly state that this work is focusing on T. congolense and T. brucei sl. only.

• There are no statistical analyses of the results. The variability of the results, that should be presented as SD not SEM, is apparently not shown in all figures. Were all tests performed at least in triplicates? These aspects should be improved.

• This work presents a very promising test, especially because it doesn’t require any pre-amplification step, yet the results could be discussed more cautiously. Comparison to other molecular tests should remain fair and the high level of criticism imposed to the other methods should also be applied to the Cas10 based 7SL-sRNA assay.

- The investigations are focusing to 2 trypanosome species only, with a limited specificity (Fig.4). The observed cross-reactivity between assays is not minimal: in the case of a mono-infection with detectable parasitemia, both Cas10 based 7SL-sRNA tests on blood will likely be positive for both species. Moreover, it would be important to further assess the specificity of the Cas10 based 7SL-sRNA assays by testing other trypanosome species, other sympatric parasites, different sample types, with host genetic material…

- The lower LoD is apparently good, yet around 1pM in most figures, not 100fM. For easier comparisons, the use of the standard parasites/ml as reference unit would be appreciated. 

- As mentioned in the discussion only, the assay is not already adapted to the field because an RNA extraction is required prior to the Cas10 assay and the commercial LFA is not adapted for field use (lack of real control band). 

In total, along the entire manuscript, it would be greatly appreciated to tone down a bit the enthusiastic promotion of this new assay that is indeed promising, but that still requires a significant work of assessment and improvement before to be usable in the field. According to the data presented in this article, it is difficult to agree that the currently proposed Cas10 based 7SL-sRNA detection assays are highly sensitive and specific. 

Minor comments

Title

• The title doesn't seem to be correct. First, the authors present a Cas10 based 7SL-sRNA detection assay for the diagnostic of active trypanosomiasis. Second, as only T. congolense and T. brucei sl. were investigated, the word ‘trypanosomiasis’ should be clarified.

Abstract

• L30. The LFA used in this study is a commercial one, hence it was not developed by the authors. 

• Please, clearly present the analytic sensitivity and specificity of the assays in the abstract with numbers.

• Please, clearly indicate that only T. congolense and T. brucei sl. were investigated.

Introduction

• L59. T. vivax can also be transmitted after cyclical development in the tsetse mouthparts.

• L63: according to the WHO reference cited, Tbg is responsible for 92% of reported cases and Tbr for 8%.

• L84-88: The list of diagnostic assays presented in this paragraph does not reflect the current diagnostic algorithm recommended by WHO. Please clearly distinguish the tests that are available and routinely used in the recommended diagnostic algorithm from the additional methods used for research, surveillance and epidemiological studies.

• L107: More than 3 trypanosome species can infect animals.

• L135-137: SHERLOCK is detecting RNA alone (or both RNA and DNA according to the strategy), but not DNA alone. Please correct this misleading statement.

• L146: A femtomolar detection level is observed for one assay in one condition shown in one figure. Hence, this statement could possibly be toned down.

Results

• Fig 5 is difficult to read. Could you please propose an alternative simpler representation? Could you use a simpler scale for the parasitemia? Could you especially highlight at what parasitemia the Cas10 assays start detecting parasites?

• In Fig 5 and 6, how do you explain the long delay before to observe positive results in the Cas10 assays? 

• From Fig 7, qPCR and Cas10 assay show similar results. In the text, it is mentioned that 4 false negative results were obtained in the Cas10 assay. How many in qPCR?

Discussion

• Do you have any idea of the stability of the excreted/secreted sRNAs in blood? How long could a 7SL-sRNA molecule remain intact and detectable in animal blood?

Reviewer #3: Grüschow and colleagues report on a Cas10 based diagnostic assay for AAT, specifically T. brucei and T. congolense. The diagnostic described here is amenable for use as both a high-throughput version and potentially, in the future, an RDT. The diagnostic is based on detection of extracellular 7SL RNA, for which the authors have already developed a qPCR based molecular diagnostic. Overall, this I find this to be a convincing assay for the diagnosis of AAT – which is needed for better treatment and surveillance of the disease. 

I have a few points/comments:

1) Given that the treatment options for T.brucei and T. congolense overlap (as far as I am aware)– why is it important to discriminate between the two for treatment? Would it not be better to be able to discriminate between T. brucei/T. congolense and T. vivax? 

2) It is stated several times that Cas10 would be a cost-effective diagnostic, could you please include a prospective costing of both a high-throughput assay and RDT in comparison to what is currently available. 

3) The benefit of this diagnostic is that, by detecting extracellular 7SL you overcome low parasite number. But would you not expect the levels of extracellular 7SL to fluctuate with the levels of parasitemia and/or be degraded over time? There seems to be some indication of this in Fig 5, where there is a slight increase in the fluorescence over time. However, in Figure 7 the fluorescent signal appears to plateau even though parasiteamia fluctuates. Could you explain why this is the case and why you anticipate long term preservation of extracellular 7SL RNA in the serum. 

5) Fluorescent 7SL-sRNA assays need to be done in triplicate and at least the SEM reported, best with the three data points plotted for each time point. Figure 2 states mean of technical duplicates; Figure 4 states SEM from 3 independent experiments, and Figures 5 and 7 read as though this is a single experiment. How reproducible is the assay, especially in serum from the cattle experiment? If it is only one assay in Figure 5 and 7 these should be repeated to at least generate triplicate data that at least error bars can be given for. 

5) Can you give the analytical sensitivity and specificity of both the Tb and Tc assays with 95 % CI. 

6) Many infections in a natural setting are most likely to be co-infections between 2 or more species. Currently, none of your assays tests the diagnostic where species are mixed. For in in vitro assay, you should do a mixing experiment of a series of proportions between Tb and Tc (ie 1:1 ; 1:5; 1:10 – 5:1; 10:1) and determine the specificity of each Cas10 assay.

PLOS authors have the option to publish the peer review history of their article (what does this mean? ). If published, this will include your full peer review and any attached files.

**Do you want your identity to be public for this peer review?** For information about this choice, including consent withdrawal, please see our Privacy Policy .

Reviewer #1: No

Reviewer #2: Yes: Brice Rotureau

Reviewer #3: No
---

## [Decision Letter · Decision Letter 1]

24 Feb 2025

Dear Career Track Fellow Grey,

We are pleased to inform you that your manuscript 'Cas10 based 7SL-sRNA diagnostic for the detection of active trypanosomosis' has been provisionally accepted for publication in PLOS Neglected Tropical Diseases.

Best regards,

Laura-Isobel McCall

Section Editor

Laura-Isobel McCall

Section Editor

Shaden Kamhawi

co-Editor-in-Chief

Paul Brindley

co-Editor-in-Chief

Reviewer's Responses to Questions

**Key Review Criteria Required for Acceptance?**

**Methods**

-Are the objectives of the study clearly articulated with a clear testable hypothesis stated?

-Is the study design appropriate to address the stated objectives?

-Is the population clearly described and appropriate for the hypothesis being tested?

-Is the sample size sufficient to ensure adequate power to address the hypothesis being tested?

-Were correct statistical analysis used to support conclusions?

-Are there concerns about ethical or regulatory requirements being met?

Reviewer #2: (No Response)

Reviewer #3: The reviewers have adequately dealt with all my comments. I have no further suggestions or comments.

**Results**

-Does the analysis presented match the analysis plan?

-Are the results clearly and completely presented?

-Are the figures (Tables, Images) of sufficient quality for clarity?

Reviewer #2: (No Response)

Reviewer #3: Analysis and Figures are acceptable

**Conclusions**

-Are the conclusions supported by the data presented?

-Are the limitations of analysis clearly described?

-Do the authors discuss how these data can be helpful to advance our understanding of the topic under study?

-Is public health relevance addressed?

Reviewer #2: (No Response)

Reviewer #3: Conclusions are supported.

**Editorial and Data Presentation Modifications?**

Reviewer #2: (No Response)

Reviewer #3: Following revision, I recommend accepting this manuscipt.

**Summary and General Comments**

Reviewer #2: The authors have carefully considered all my comments. They have either modified the manuscript accordingly or clearly discussed several aspects that were not clear for me.

Reviewer #3: No further comments

PLOS authors have the option to publish the peer review history of their article (what does this mean? ). If published, this will include your full peer review and any attached files.

**Do you want your identity to be public for this peer review?** For information about this choice, including consent withdrawal, please see our Privacy Policy .

Reviewer #2: **Yes: ** Brice Rotureau

Reviewer #3: No

---

## [Editor Report · Acceptance letter]

Dear Career Track Fellow Grey,

We are delighted to inform you that your manuscript, "Cas10 based 7SL-sRNA diagnostic for the detection of active trypanosomosis," has been formally accepted for publication in PLOS Neglected Tropical Diseases.

Best regards,

Shaden Kamhawi

co-Editor-in-Chief

Paul Brindley

co-Editor-in-Chief
